# Feasibility characteristics of wrist-worn fitness trackers in health status monitoring for post-COVID patients in remote and rural areas

**Madeleine Wiebe**[1]*, **Marnie Mackay**[2], **Ragur Krishnan**[3], **Julie Tian**[2], **Jakob Larsson**[1], **Setayesh Modanloo**[2], **Christiane Job McIntosh**[4], **Melissa Sztym**[5], **Gail Elton-Smith**[5], **Alyssa Rose**[5], **Chester Ho**[6,7], **Andrew Greenshaw**[2], **Bo Cao**[2], **Andrew Chan**[6], **Jake Hayward**[8]

1 Faculty of Medicine and Dentistry, University of Alberta, Edmonton, Canada, 2 Department of Psychiatry, Faculty of Medicine and Dentistry, University of Alberta, Edmonton, Canada, 3 Department of Electrical and Computer Engineering, Faculty of Engineering University of Alberta, Edmonton, Canada, 4 Neurosciences, Rehabilitation and Vision Strategic Clinical Network, Alberta Health Services, Edmonton, Canada, 5 Covenant Health Rural Health Services, Edmonton, Canada, 6 Glenrose Rehabilitation Hospital, Edmonton, Canada, 7 Department of Physical Medicine and Rehabilitation, Faculty of Medicine and Dentistry, University of Alberta, Edmonton, Canada, 8 Department of Emergency Medicine, Faculty of Medicine and Dentistry, University of Alberta, Edmonton, Canada

* mcwiebe@ualberta.ca

**Data Availability Statement:** All relevant data are within the manuscript and its Supporting Information files.

## Abstract

### Introduction

Common, consumer-grade biosensors mounted on fitness trackers and smartwatches can measure an array of biometrics that have potential utility in post-discharge medical monitoring, especially in remote/rural communities. The feasibility characteristics for wrist-worn biosensors are poorly described for post-COVID conditions and rural populations.

### Methods

We prospectively recruited patients in rural communities who were enrolled in an at-home rehabilitation program for post-COVID conditions. They were asked to wear a FitBit Charge 2 device and biosensor parameters were analyzed [e.g. heart rate, sleep, and activity]. Electronic patient reported outcome measures [E-PROMS] for mental [bi-weekly] and physical [daily] symptoms were collected using SMS text or email [per patient preference]. Exit surveys and interviews evaluated the patient experience.

### Results

Ten patients were observed for an average of 58 days and half [N = 5] were monitored for 8 weeks or more. Five patients [50%] had been hospitalized with COVID [mean stay = 41 days] and 4 [36%] had required mechanical ventilation. As baseline, patients had moderate to severe levels of anxiety, depression, and stress; fatigue and shortness of breath were the most prevalent physical symptoms. Four patients [40%] already owned a smartwatch. In total, 575 patient days of patient monitoring occurred across 10 patients. Biosensor data was usable for 91.3% of study hours and surveys were completed 82.1% and 78.7% of the

**Funding:** The author(s) received no specific funding for this work.

**Competing interests:** The authors have declared that no competing interests exist.

time for physical and mental symptoms, respectively. Positive correlations were observed between stress and resting heart rate [r = 0.360, p<0.01], stress and daily steps [r = 0.335, p<0.01], and anxiety and daily steps [r = 0.289, p<0.01]. There was a trend toward negative correlation between sleep time and physical symptom burden [r = -0.211, p = 0.05]. Patients reported an overall positive experience and identified the potential for wearable devices to improve medical safety and access to care. Concerns around data privacy/security were infrequent.

## Conclusions

We report excellent feasibility characteristics for wrist-worn biosensors and e-PROMS as a possible substrate for multi-modal disease tracking in post-COVID conditions. Adapting consumer-grade wearables for medical use and scalable remote patient monitoring holds great potential.

### Author summary

Consumer-grade wearables, such as wrist-worn devices, are able to measure several health parameters including step count and heart rate, increasing their potential for monitoring patients from rural locations or with restricted access to health care. Patients recovering from COVID-19 in rural communities were recruited and to wear FitBit Charge 2 devices as much as possible during the study, in order to collect health parameters. In addition to this, patients completed daily and weekly surveys focusing on mental and physical health, and an exit interview with researchers describing their overall experience with the devices. Overall, increasing stress levels was associated with increased heart rate, anxiety, and interestingly daily steps. Decreased sleep time was observed to be associated with increased burden of symptoms. As a group, patients reported an overall positive experience during the study and felt that the devices increased their access to care and contributed to recovery and safety. A small number of patients shared concerns with data privacy and security with the devices. In summary, these devices are very feasible to use for tracking symptoms and health outcomes in rural patients recovering from infection.

## Introduction

Recovery from COVID can be long and unpredictable. Some patients return to premorbid health quickly while others endure a debilitating and poorly understood recovery path lasting months to years, termed 'long-COVID', comprising both mental and physical symptoms [reviewed in 1,2]. Traditional, clinic-based methods of disease assessment are episodic, limiting their utility in a complex and undulating disease course such as post-COVID. New approaches are needed that can track longitudinal physiologic changes at the individual patient level.

Wearable devices may be ideal tools for studying complex diseases and post-COVID conditions [reviewed in 3]. Biosensor metrics have been shown able to predict health outcomes for a range of chronic diseases including congestive heart failure, COPD, hypertension, diabetes, and more [4–7]. More recently, smartwatches have been used to detect pre-symptomatic COVID infections [8]. Common and affordable smartwatches and fitness trackers might

support remote patient monitoring (RPM) if the emergent data is high-enough quality to inform safe decision making [9–12]. Remote patient monitoring is particularly important for patients living in remote and rural areas where in-person care can be limited [13–15].

Feasibility data for wearable devices varies with population, technology, and monitoring protocol and little has been published for post-COVID conditions, especially in rural settings. The objective of our study was to describe the feasibility of using common, consumer-grade wrist-worn biosensors (FitBit fitness trackers) for disease parameter tracking in post-COVID conditions in a rural community, including both technical aspects (e.g. data quality/completeness), usability, and the overall patient experience using the devise during recovery. We hypothesized that smartwatches can deliver clinically useful, passively collected biometric data for this unique population.

## Materials and methods

### Patient cohort

Patients were recruited from an early supported discharge (ESD) program in Camrose, Alberta, Canada, which has a population of just under 21,000 residents. The ESD program was originally designed to support patients recovering from acute stroke [16]; however, during the COVID-19 pandemic, it was adapted for COVID-19. Patients were referred to the program either directly from hospital (at discharge) or through community clinics. The ESD team is a rehabilitation team that delivers intensive rehabilitation programs in the home using both in-person visits and telemedicine (video or phone). The team typically includes occupational therapy, physical therapy, speech-language pathology, social work, nursing, therapy assistants, recreation therapy and a psychologist. Patients were also recruited through a virtual post-COVID pilot program and were located throughout Alberta, primarily in rural settings. Patients were referred to this program through the Rehabilitation Advice Line. Rehabilitation in this program was provided solely via telemedicine.

First, patients were enrolled in the ESD program by the clinical team (nurses, physiotherapists, occupational therapists) using standard program criteria: 18 years of age or older, ambulatory, with mild to moderate functional impairment and sufficient cognitive capacity to participate in the program. Researchers were not involved at this stage. Then, all enrolled ESD patients were screened for study eligibility by the research team. Patients who owned a smartphone and were capable of completing surveys were included, regardless of medical comorbidity including substance use disorders. Excluded patients were those without internet connectivity, who were unable to complete surveys (ex. language and speech deficits, non-English-speaking), or for whom device retrieval was deemed to be unlikely by the study investigators.

### Data sources

Data sources included: 1) wearable devices, 2) health records, 3) digital surveys, and 4) patient interviews. Health record data comprised diagnostic, medication, and procedure codes occurring in the year before enrollment and during the observation period.

### Devices

FitBit Charge 2 devices, which are considered consumer-grade wearables, were provided to each patient at enrollment. The FitBit Charge 2 is a wrist-worn fitness tracker that includes a triaxial accelerometer, an altimeter, and an optical heart rate tracker. Parameters collected included activity (steps per day), heart rate (beats per minute), and duration (minutes) and

quality (% deep) of sleep. Data was sent via Bluetooth to a patient-owned smartphone, uploaded to the FitBit cloud database and then extracted through the third-party platform (Fitabase; https://www.fitabase.com/).

## Monitoring protocol

Patients received an orientation to the device on study enrollment and were instructed to wear it as much as possible, up to 24 hrs a day (except when charging). There were no specific instructions on how to use device data and patients were free to share data with the clinical team if desired; however, data was not directly transmitted to the clinical team. If the patient had questions about the device or technical issues, a research nurse was available for support by phone during daytime hours. Heart rate was collected minute-by-minute and averaged over the day. Step counts were summations of the day.

## Surveys

Patients received digital surveys via SMS text or email (per patient preference) using the RedCap [17] survey platform. Baseline health status surveys included measures of mental and physical health and a technology experience survey (Comprising two sections: Tech comfort [7-items, avg score [0–5]], Health Literacy [4-items, avg score [0–5]] adapted from the Telehealth Usability Questionnaire [TUQ] [18]. During the follow-up period physical and mental symptom surveys were delivered daily and bi-weekly, respectively. English versions of validated self-reported screening scales were used to measure severity of stress, anxiety, and depression symptoms, including the Perceived Stress Scale (PSS), a 10-item questionnaire with a Cronbach's alpha of >0.70 used to assess level of stress in the previous month (PSS; PSS score ≥14 indicates moderate or high stress [19]; the Generalized Anxiety Disorder 7-item (GAD-7) scale, a 7-item questionnaire with a Cronbach's alpha of 0.92 and used to assess the self-reported levels of anxiety in respondents in the two weeks prior to assessment (GAD-7 score ≥10 indicates likely generalized anxiety disorder [GAD]) [20]; and the Patient Health Questionnaire-9 (PHQ-9) a 9-item questionnaire with a Cronbach's alpha of 0.89 and used to assess the severity of depression symptoms (for PHQ-9; a score ≥10 indicates likely major depressive disorder (MDD)) [21].

Detailed description of physical symptom surveys and results can be found in the supplementary materials, S1 Table. In short, 18 total symptom scales (6-point scale, 'absent' [1] to 'very severe' [6]) were grouped by body system (constitutional, gastrointestinal, neurologic, respiratory, cardiovascular). At discharge, patients received an exit survey that included feedback on the devices, also adapted from the TUQ.

## Interviews

To examine the mechanisms that relate the outcomes to the potential predictors and to generate contextualized information about processes related to remote patient monitoring and consumer-grade wearable devices, we used mixed-methods with qualitative analysis of interviews integrated with our quantitative data analysis. At the conclusion of the study patients were approached for interviews. Questions focused on technology usability, acceptance, and perceived barriers to use, such as data privacy/security. Author MW completed interviews virtually using Zoom and curated auto-generated transcriptions.

## Study outcomes

**Primary outcome: Protocol adherence.** Our primary outcome was protocol adherence, defined as device wear time (% of minutes with analyzable heart rate data) and survey

completion rates. Patient interviews and survey responses added context for this outcome, exploring patient perceptions and experience.

**Secondary outcomes: Biosensor parameter correlations.** Secondary outcomes were associations between 1) the primary outcome (adherence) and patient characteristics (e.g., age, sex, symptom severity, technology readiness measures, and time under observation), and 2) device biometrics (activity/HR/sleep) and disease outcomes (physical and mental symptoms).

### Statistical analysis

Descriptive statistics (mean and standard deviation [SD]) were calculated for primary and secondary outcomes using Microsoft Excel. We then conducted exploratory analyses of potential associations within and between participants on biometric parameters (no power calculations were performed). First, we performed a within-subject longitudinal analysis using paired t-tests to evaluate for changes in symptom severity over time in the subset of subjects observed for more than 8 weeks [N = 5], comparing weeks 1–4 and weeks 8+. We then used Pearson correlation coefficients to evaluate group-level trends in weekly aggregated data.

### Qualitative analysis

We conducted thematic analysis of interview transcripts using a combined deductive and inductive approach, aided with NVIVO software. Two team members (MW and JL) reviewed original transcripts to identify themes of interest and compared results. Where there was disagreement, consensus was obtained through discussion. Smaller themes were sequentially grouped into larger categories until core themes were identified.

### Funding and Ethics

This study was registered and approved by the University of Alberta Research Ethics Board [Pro00113943].

## Results

### Participant characteristics

Between November 2021 and May 2022, 18 patients were approached for enrollment; three declined and one patient was ineligible due to lack of wireless connectivity. After enrollment, 3 of 14 patients withdrew consent, and 1 was removed due to technical issues with their device, leaving 10 patients for the final analysis (as seen in Fig 1). Of the patients who withdrew consent, one felt overwhelmed with the added demands of the device in addition to their rehabilitation, one preferred to use their own smartwatch, and one shared concern about data sharing and security. Seven patients consented to an interview. The overall recruitment rate was therefore 61.1%.

Table 1 shows baseline patient characteristics. Mean age was 53 years (SD 15.0) and 80% (8/10) of patients were female. Five (50%) had been hospitalized with COVID (average length of stay = 41 days) and four (40%) had required ventilatory support in the ICU. The remaining 50% were referred from community clinics. On average, patients had fewer than one documented medical condition prior to their COVID-19 diagnosis and half (5/10) owned their own smartwatch. The most common physical symptoms were 'constitutional' (ex. fatigue, poor sleep), followed by 'respiratory' and 'cardiovascular'. Average mental health symptoms scores were GAD-7 = 15.7 (severe anxiety), PHQ-9 = 14.8 (moderate depression) and PSS = 22.7 (moderate stress). Patients had moderate to high levels of familiarity with technology (ex. comfort with technology = 3.6/5; health literacy = 3.7/5). Detailed patient

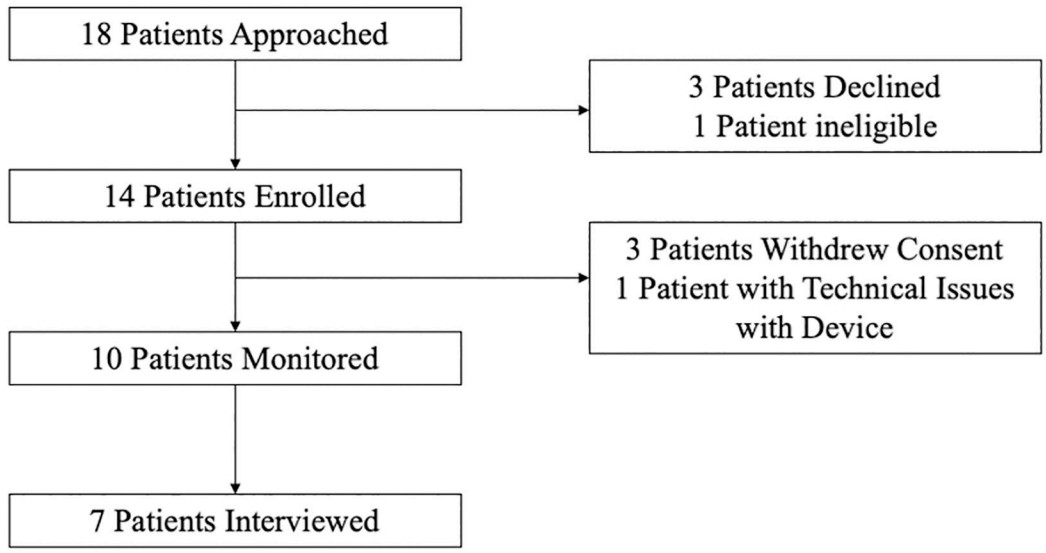

**Fig 1. Flow diagram of study recruitment.**

**Table 1. Baseline patient characteristics [N = 10].**

| Demographics | |
|---|---|
| Age (mean [SD]) | 53 [15] |
| Female Sex no. [%] | 8 [80%] |
| Hospitalized no. [%] | 5 [50%] |
| Days in hospital (mean [SD]) * | 41.0 [21.1] |
| ICU stay no. [%] | 4 [40%] |
| Ventilator no. [%] | 4 [40%] |
| Number of days in ESD (mean [SD]) | 58 [117] |
| Home oxygen [%] | 1 [10%] |
| Comorbidities per patient (mean [SD]) | 0.8 [0.8] |
| Medications per patient (mean [SD]) | 4.4 [3.1] |
| **Familiarity with Technology** | |
| Already own a smartwatch [%] | 4 [40%] |
| Technology comfort (7-items, mean score [0–5]) | 3.6 [1.4] |
| Health Literacy (4-items, mean score [0–5]) | 3.7 [1.1] |
| **Physical symptoms (mean score [0–4])** | |
| Constitutional (6 symptoms) | 1.43 [0.83] |
| Gastrointestinal (3 symptoms) | 0.31 [0.55] |
| Neurological (3 symptoms) | 0.61 [0.62] |
| Respiratory (4 symptoms) | 0.90 [0.55] |
| Cardiovascular (2 symptoms) | 0.68 [0.90] |
| **Mental health symptoms (mean score [SD])** | |
| General Anxiety Disorder-7 (out of 21) | 15.7 [1.3] |
| Patient Health Questionnaire-9 (out of 27) | 14.8 [1.3] |
| Perceived Stress Scale (out of 40) | 22.7 [1.4] |

* Hospitalized patients [N = 5]

**Table 2. Reported health parameters and survey outcomes across the clinical course.**

| Category | Weeks 1–4 [N = 10] Average [SD] | Weeks 4–8 [N = 9] Average [SD] | Weeks 8+ [N = 5] Average [SD] |
|---|---|---|---|
| **% Wear time (minutes with heart rate data)** | 91.5 [5.4] | 92.9 [4.4] | 88.1 [5.7] |
| **Sleep (hours)** | 5.7 [2.3] | 7.0 [2.0] | 5.6 [3.2] |
| **Deep sleep [%]** | 13.3 [10.0] | 17.4 [7.5] | 15.9 [8.2] |
| **Resting heart rate** | 68.7 [7.6] | 66.1 [9.5] | 66.7 [8.9] |
| **% Completion for surveys** | | | |
| **Physical** | 82.8 [12.3] | 81.1 [18.1] | 82.5 [11.2] |
| **Mental** | 92.1 [15.3] | 81.0 [23.1] | 47.9 [36.1] |
| **Physical symptoms (mean [SD]; range 0–4)** | | | |
| **All** | 0.8 [0.6] | 0.7 [0.5] | 0.7 [0.4] |
| **Constitutional** | 1.0 [0.8] | 0.9 [0.8] | 1.1 [0.7] |
| **Neuro** | 0.6 [0.6] | 0.6 [0.6] | 0.8 [0.8] |
| **Gastrointestinal** | 0.3 [0.6] | 0.2 [0.5] | 0.3 [0.5] |
| **Respiratory** | 0.9 [0.6] | 0.8 [0.6] | 0.8 [0.5] |
| **Cardiovascular** | 0.7 [0.9] | 0.6 [0.9] | 0.3 [0.6] |
| **Mental health symptoms** | | | |
| **Mood (out of 27)** | 20.0 [8.0] | 19.3 [9.0] | 16.4 [7.8] |
| **Anxiety (out of 21)** | 16.2 [6.8] | 15.8 [6.8] | 12.4 [6.4] |
| **Stress (out of 40)** | 30.5 [6.1] | 28.5 [4.9] | 26.6 [5.5] |
| **Activity: Steps** | | | |
| **Daily count** | 3042 [1606] | 4330 [3338] | 3284 [2249] |
| **Activity: Intensity (minutes/day)** | | | |
| **Low** | 127.3 [30.2] | 156.6 [71.7] | 121.6 [74.3] |
| **Moderate** | 6.4 [6.6] | 12.6 [16.9] | 12.4 [20.3] |
| **High** | 4.8 [6.2] | 5.2 [6.6] | 5.6 [6.1] |

characteristics, including comorbidities and medications, are shown in Supplementary S2 Table, S1 Fig and S2 Fig.

## Descriptive analysis

Table 2 summarizes descriptive statistics for primary adherence outcomes (group aggregate). Overall, 575 patient days of patient monitoring occurred across 10 patients, patients were observed for 58 days on average and half [N = 5] were monitored for 8 weeks or more. For the primary outcome (adherence), heart rate data was available for 88.1–91.5% of study hours and physical and mental symptom surveys were completed 81.1–82.8% and 47.9–92.1% of the time, respectively, depending on week of observation. On all measures, adherence decreased as the study progressed. At baseline, physical symptom burden was mild (average rating of 0.8/4) with constitutional and respiratory symptoms being the most severe. Average group activity, sleep duration and quality (proportion time in deep sleep) were highest in weeks 4–8. Average resting heart rate was relatively constant over time.

## Within and between participant associations

S2 Table shows within-participant changes in biometric parameters over time for the 5 participants observed for more than 8 weeks. Except for respiratory symptoms (mean difference -0.23, p = 0.03), temporal changes did not reach statistical significance with this limited sample size. Parameters showing notable trends included gastrointestinal (-0.03,

**Table 3. Correlation coefficients between FitBit data outputs and patient symptoms.**

| Survey parameter | Resting Heart Rate | Steps | Sleep Time |
|---|---|---|---|
| Physical Symptoms | 0.011 [P = 0.920] | 0.116 [P = 0.30] | -0.211 [P = 0.05] |
| Anxiety [GAD-7] | 0.205 [P = 0.070] | **0.289 [P<0.01]** | -0.186 [P = 0.09] |
| Mood [PHQ-9] | 0.148 [P = 0.189] | 0.183 [P = 0.10] | 0.004 [P = 0.97] |
| Stress [PSS] | **0.360 [P<0.01]** | **0.335 [P<0.01]** | 0.208 [P = 0.06] |

p = 0.17) and cardiovascular symptoms (-0.39, p = 0.09), total steps (-108.4) and average heart rate (-5.99, p = 0.10). Survey response rates were correlated with device wear time (r = 0.67, p = 0.03), however there were no statistically significant associations between wear-time and other patient characteristics (for example, demographics, symptoms, previous technology experience or time under observation; S3 Table). Table 3 shows group-level associations between biosensor metrics [weekly aggregate]. Resting heart rate was positively correlated with stress [**r = 0.360, p<0.01**]; step count was correlated with anxiety [**r = 0.289, p<0.01**] and stress [**r = 0.335, p<0.01**]. Correlations for sleep time were not statistically significant, however trends suggested negative correlations with physical symptoms [r = -0.211, p = 0.05] and anxiety [r = -0.186, p = 0.09] and a positive correlation with stress [r = 0.208, p0.06].

## Individual sensor data: case examples

Fig 2 shows data for two patients, illustrating the complexity and heterogeneity of individual data. At enrollment, Patient A experienced marked physical symptoms, a high resting heart rate (79–83 bpm) and low daily step counts (approximately 1km). Over time, their physical symptoms improved, resting heart rate decreased, and daily activity increased. In contrast,

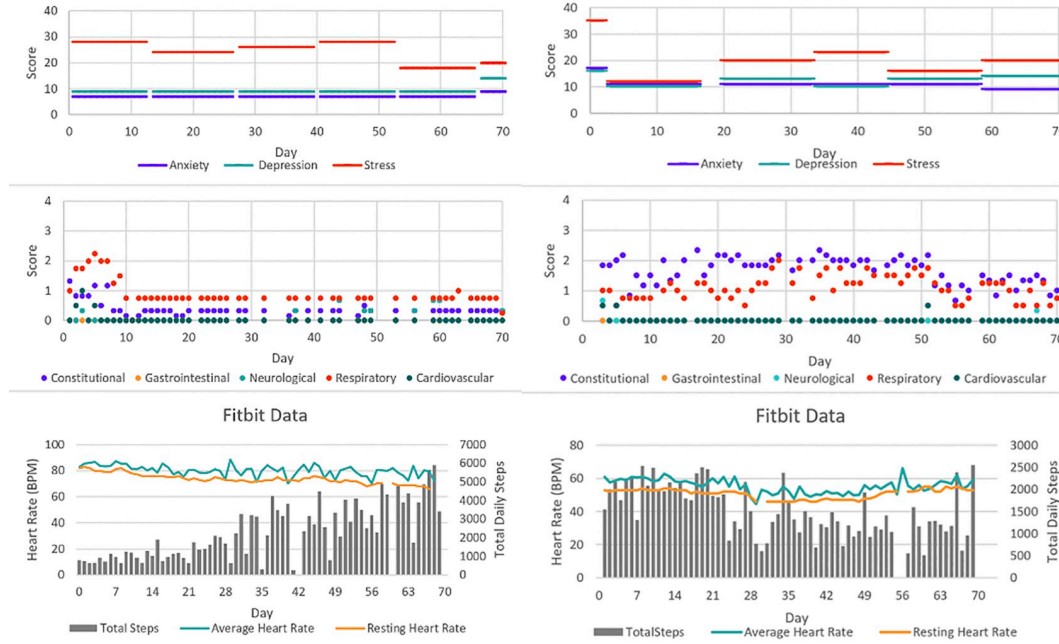

**Fig 2.** Sample longitudinal data for Patient A (left) and B (right), including results from symptom surveys, heart rate and total daily number of steps.

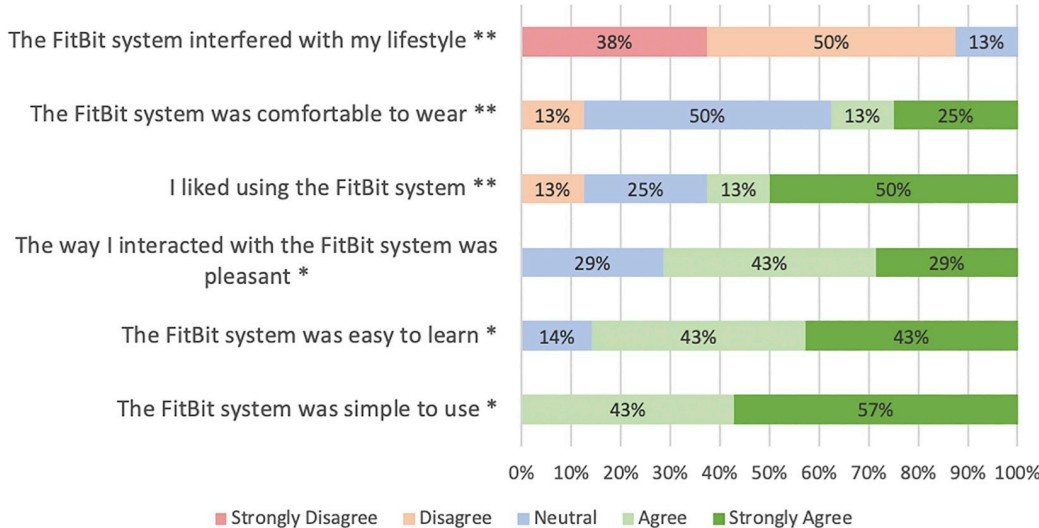

**Fig 3. Survey results for questions pertaining to device acceptability with answers ranging from strongly disagree to strongly agree.** *7 out of 10 participants responded; **8 out of 10 participants responded.

Patient B's physical symptoms remained prominent throughout observation. Their resting heart rate was lower than Patient A (46–54 bpm) and didn't significantly change over time. Their activity levels didn't show a clear temporal change. Mental health symptoms were prominent in both patients.

## Surveys results

Figs 2 and 3 display survey results; additional survey data is presented in supplementary materials (S1 Fig). Proportions herein represent overall levels of agreement (agree or strongly agree) vs. disagreement [disagree or strongly disagree], as seen in Fig 3. Of those who responded, the majority liked using the FitBit system (5/8 [63%]), found it easy to learn (6/8 [86%]) and simple to use (7/7 [100%]); no patients found the device interfered with their lifestyle and only one (13%) found the device uncomfortable. Three-quarters (6/8 [75%]) of the patients found that wearing a device made them feel safer and 4/8 (50%) felt it helped them to better understand their disease [Fig 4]. A minority (2/8 [25%]) used the device to help decide when to seek medical care and no patients (0/8) reported that the devices caused anxiety.

## Theme analysis of interview transcript

Two male and five female patients completed exit interviews. Key themes are summarized below, and representative quotes are presented in Table 4.

**Support in isolated environments.** Most felt that wearable devices are likely to improve medical safety for rural communities in the future, possibly through symptom, activity, and vital sign data transmission to clinical teams, family members and caregivers.

**Gaining disease insights.** Some patients spontaneously use device data (mostly heart rate) to help them interpret physical symptoms, even without medical guidance. The impacts of biosensor data on mental health were varied. Some found that normal heart rate readings reassured them while others found that elevated heart rates worsened anxiety about their disease.

**Concerns around privacy.** Few were concerned data privacy and security. One participant was hesitant to share their data with for-profit device companies, specifically.

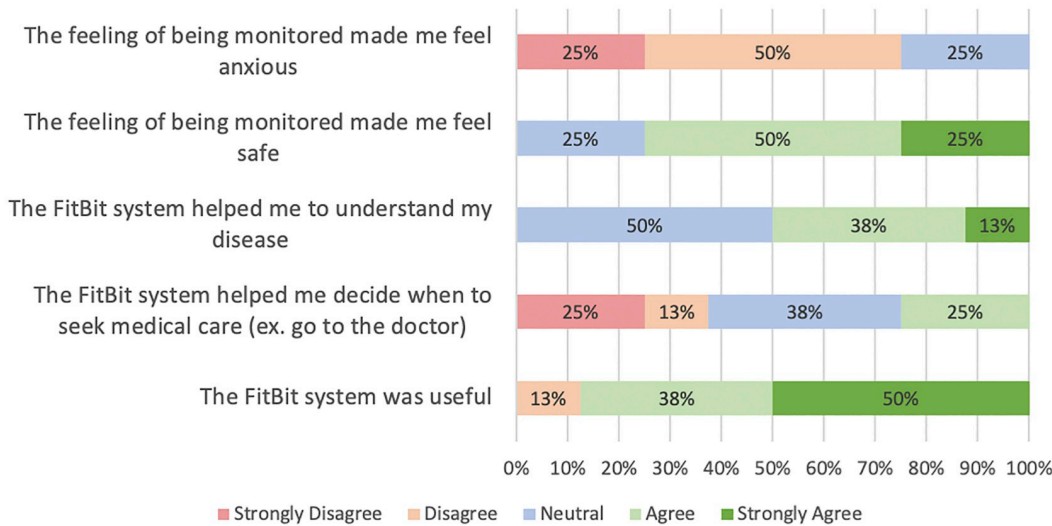

**Fig 4. Survey Results for questions pertaining to psychological impact and clinical use with answers ranging from strongly disagree to strongly agree.**

**Potential for research.** Almost universally, participants felt wearable devices held great promise as tools for clinical research and expressed a willingness to contribute their data to the research community. Participants were motivated by a desire to see advancements in understanding and treatment of post-COVID conditions.

## Discussion

We report excellent feasibility characteristics for wrist-worn fitness trackers in post-COVID disease monitoring in rural areas, including hypothesis generating associations suggesting that biosensor data can deliver disease insights at the individual level. Biosensor data was complex and heterogenous, highlighting the need for machine learning techniques to separate signal from noise. For instance, cutting edge time-series analysis and signal detection techniques might enable patient-level detection of meaningful change in biometrics that could suggest new or worsening disease [8]. Our qualitative work illustrated a unique lived experience for rural patients with technology and unmet needs for remote monitoring tools that integrate with existing clinical care models to ease transitions from hospital to home.

Compared to existing studies, we observed high levels of protocol adherence. For common disease targets like COPD and CHF, RPM protocol adherence is typically low and decreases over time [7,22–23]. Our post-COVID population was younger than most existing RPM cohorts, had relatively few comorbidities and were generally comfortable with technology, perhaps explaining our favorable result. Also, most existing studies RPM protocols use medical-grade devices, which are often unfamiliar to patients and challenging to operate. Consumer choice in technology is key driver of success for RPM and consumer-grade devices that offer non-medical applications present an opportunistic substrate for building scalable RPM platforms [24]. The drawback of consumer-grade biosensors is that sensor accuracy is often unknown and cannot be assumed clinical grade [25]. However, as sensor accuracy and signal detection algorithms improve for longitudinal patient data, we will be able to better compensate for imperfection in underlying discrete data. Learning to process data from consumer-grade biosensors is foundational for just-in-time medical decision making, especially in

**Table 4. Qualitative analysis of exit interviews with sample quotes from participants.**

| Common themes of interviews | # Of quotes | # Of participants | Sample participant narratives | |
|---|---|---|---|---|
| | | | Examples of device use addressing needs of patient | Examples of device unable to address needs of patient |
| Support in isolated or rural environments | 5 | 3 | "Well, it was good because I knew that if I was [inputting information], and somebody at the other end was seeing that something was really wrong they would have notified the girls in Camrose and would have notified medical assistance immediately. So, it was kind of another area of peace of mind. . .yeah, we don't have next door neighbors. We can't run for help. You know, we're sort of isolated out here, so like I say, we're on our own. You kind of just grasp at things, and you watch for signs. But you're not always fully aware, maybe, of what's happening until it's too late" [2/7 participants] | "I don't know, I didn't get a full view on what all [the device] could do, simply because it was kind of one-sided. I knew the information was going out, but I didn't necessarily know what was happening with it. I'm not having that feedback from a medical professional. I feel like it would be a really important piece if this is being used in the future." [1/7 participants] |
| Improved insight into symptoms | 25 | 7 | ". . .I felt like I've gone crazy for a long time, and it just helps me, even if it's not my heart rate, and it's something else. It's just a peace of mind that it, you know, it's not my heart." "I quite like the monitoring aspect from home, just because it gives me a bit more of an insight of what my heart rate is and what it's doing; just a better overall understanding." [6/7 participants] | "I think, because I was having so many problems with my heart rate, and then I was tracking it so closely, it did make me hyper-aware of my heart rate a little bit. . .because now I was really closely monitoring, and it wasn't going well. So then, I was really concerned about that. [2/7 participants] |
| Concern for privacy | 7 | 7 | "None whatsoever. Like I said, I want to figure out what's going on, and I don't want somebody else to have to go through what I'm already dealing with. They might as well use me to figure out what we can do for others, right? So, I have no concerns with it whatsoever." [6/7 participants] | "It would again come down to what the requirements were for [data sharing]. . . . or at the very least had some transparency about what they were doing with it. . .my information is out and I don't know what it's taking off of my phone. What sort of data it's gathering on me. So that's something that I see as a limitation of health monitoring." [1/7 participants] |
| Research potential of devices | 8 | 6 | "Well, I think it was helpful for whoever was tracking to get a better insight into what the aftereffects of COVID does, everybody's different. It's a very needed program and when you have somebody who went through what [my spouse] did, it's really helpful. Maybe it'll help someone else in [the ESD program]. I think the program is very much needed." [5/7 participants] | "I've heard of studies where because of where the money is coming from, there's a risk of results in the data being made to fit the needs of the corporation or person or group or whatever. If the data is being used by people who are motivated financially rather than scientifically, and not with the goals of the health of individuals and people." [1/7 participants] |

relatively healthy patient populations who haven't been prescribed a medical grade device [26]. Our study underscores the need to develop patient-centered technologies built with the non-medical consumer in mind.

Our group level associations and individual case examples highlight the potential for smart-watches to deliver novel disease insights in scientific studies. In our post-COVID patients, anxiety and stress correlated with activity and resting heart rate while physical symptom correlated with sleep. This granular, continuous data paints a rich picture of disease at the individual level, forming a foundation for precision medicine and improving on traditional forms of episodic measurement. Given our small sample size, our observed group-level associations are hypothesis generating only. Subjectively, the heterogeneity seen across individual data supports the hypothesis that post-COVID conditions comprise a range of distinct phenotypes rather than a single entity [27–28], a possibility that will hopefully be further elucidated in ongoing clinical trials that utilize wearable devices [29]. As more studies using wearable bio-sensors emerge it will be important to create and adhere to standards in metric derivation and validation so that results are clinically applicable.

Our patient interviews provided a rich context for interpreting quantitative data and brought to the fore a range of patient experiences that are key for understanding the future

impacts of wearable devices in medicine. Continuous physiologic data can amplify or dampen anxiety around disease depending on the individual. In some instances, normal device data helped to reassure when symptoms were fluctuating unpredictably. For others, abnormal heart rate data worsened anxiety around mild or absent physical sensations. The potential impacts of personalized biodata on mental health are poorly described in existing literature and for some it could contribute to excess healthcare utilization [30]. Our result highlights the need to better understand these impacts through further research and to improve medical support patients who choose to wear devices. Interestingly, concerns about data-sharing and privacy were infrequently raised by our patients but are known to be major determinants of success for wearable devices in healthcare, overall [31–32].

Patients unfamiliar with wrist-worn devices in our study required a significant amount of time and support during device set-up. For those who owned a device already, changing to a new one was unpopular, as expressed during enrollment with some withdrawing consent, and in the post-study interviews among those who had previously owned a different device. Human factors like this are critical determinants of success for RPM platforms. Shin et al. [33] found that wearables are best accepted by patients who are motivated to monitor a chronic condition. Yin et al. [34] reported that convenience, social influence, and expectation of improved health are important for uptake, while cost and perceived risk are less so. In rural settings, such as ours, perceived benefits for wearable technologies might be elevated as in-person services are often inaccessible [reviewed in 35]. In future work using smartwatches rather than fitness trackers would likely yield even stronger feasibility data given the attractiveness of their non-medical applications like text messaging, music, and social networking, all of which encourage wear-time. Indeed, many patients in our study were particularly reluctant to use a fitness tracker if they already owned a smartwatch.

Technological literacy is major determinant of success for wearable devices. In our study, some patients chose not to participate because they were not comfortable with technology. For those who enrolled, the ESD team provided technical support throughout the study. Thus, our feasibility results are specific to those enrolled and to the ESD environment, and while we observed positive results, a low recruitment rate (about 60%) suggests that there remains a large, untapped potential for digital tools in post-COVID patients. Future work should aim to explore reasons for study drop-out or non-enrollment and determine if added education or support, or more customized technologies might increase the reach of these interventions.

## Limitations

Our study has several limitations. Most notably, our sample size was small, and it is unclear if our findings can be replicated a larger population. Slow recruitment was due to multiple factors. Primarily, the ESD program only treats a subset of post-COVID patients with mild-moderate disability, excluding the most mild and severe cases. As conditions changed during our study (vaccines emerged and prevalent viral strains shifted) we observed that the post-COVID phenotype also changed. In the early stages of our work, we observed more severe respiratory symptoms and most patients had been hospitalized, often recovering from a period of mechanical ventilation. In the later stages, patients were often referred from community clinics with more mild respiratory disease but more severe constitutional symptoms (ex. fatigue or poor sleep). This progression to more mild disease slowed enrollment in the ESD program and therefore slowed study recruitment. Also, the shifting disease phenotype makes it challenging to generalize our results to future endemic states for COVID.

Importantly, in this setting our patients were visited near-daily in their homes or via telemedicine by therapists, which could have artificially increased protocol adherence, and critically, a relatively low recruitment rate (60 percent) suggests that population-level impacts for

wearable technologies remains uncertain. It is also critical to note that consumer-grade devices designed around healthy individuals could be error-prone in patients with disease, thus relative changes in metrics should be considered. We chose percent of minutes with analyzable heart rate data as a measure of device wear-time and as a surrogate measure of protocol adherence. Changes in device positioning, fit, or environmental interference could affect or interrupt device readings even when the device is worn continuously. Ultimately, we observed high adherence and data usability throughout the study, suggesting that these unmeasured factors minimally affected our result. Also, from an analytic standpoint, real-world adherence including incorrect use and environmental factors is the most important outcome.

Additionally, as mentioned by a participant in the post-study interviews, device companies have the potential to change device algorithms and collect data without the user knowing. Discomfort with data sharing was also a reason one participant withdrew from the study. Also, generalization to other devices is difficult due to the swift technological advancements and differences between companies and models. Of the three patients who withdrew consent, one noted that they preferred a different device brand, a theme that was also noted in the exit interview of those who participated.

## Conclusions

We find promising feasible characteristics for wrist-worn devices in remote disease tracking for post-COVID conditions in rural communities. Our data are foundational for future testing of consumer-grade devices in the medical sphere, highlighting the need for co-created RPM platforms designed with patient/consumer technology preference in mind. Such innovations have great potential to improve healthcare access and safety for otherwise isolated populations.

## Supporting information

**S1 Table. Mean difference in average symptoms scores for weeks 1–4 vs. 8+ for patients participating for more than 8 weeks [N = 5].**
(DOCX)

**S2 Table. Patient characteristics–complications for hospitalized patients [N = 5], comorbid conditions captured with administrative data diagnostic codes [N = 10], medications in the year prior to COVID-19 diagnosis generated by WHO ATC coding system, and symptom scores by body system.**
(DOCX)

**S3 Table. Correlation coefficients between device wear-time [pooled weekly] and patient characteristics, symptom severity, technology acceptance and time under observation [N = 10].**
(DOCX)

**S1 Fig. Baseline technology comfort survey responses [N = 10].**
(EPS)

**S2 Fig. Baseline health literacy survey responses [N = 10].**
(EPS)

## Acknowledgments

We acknowledge the extensive efforts of the Camrose Early Supported Discharge team (Dana Norton, Alyssa Rose, Sharene Lamson) in providing clinical and administrative support. We

also acknowledge the contributions of the Glenrose research team (Elton Lam, Tod Vandenberg, Stephan Pham) for their provision of analytic support.

## Author Contributions

**Conceptualization:** Chester Ho, Andrew Greenshaw, Jake Hayward.

**Data curation:** Madeleine Wiebe, Marnie Mackay, Jakob Larsson, Jake Hayward.

**Formal analysis:** Ragur Krishnan, Julie Tian, Jakob Larsson, Setayesh Modanloo, Jake Hayward.

**Funding acquisition:** Chester Ho, Andrew Greenshaw, Jake Hayward.

**Investigation:** Madeleine Wiebe, Jakob Larsson, Setayesh Modanloo, Jake Hayward.

**Methodology:** Jake Hayward.

**Project administration:** Madeleine Wiebe, Marnie Mackay, Jake Hayward.

**Resources:** Christiane Job McIntosh, Melissa Sztym, Gail Elton-Smith, Alyssa Rose, Jake Hayward.

**Software:** Madeleine Wiebe, Jake Hayward.

**Supervision:** Chester Ho, Andrew Greenshaw, Bo Cao, Andrew Chan, Jake Hayward.

**Validation:** Jake Hayward.

**Visualization:** Julie Tian, Jake Hayward.

**Writing – original draft:** Madeleine Wiebe, Jake Hayward.

**Writing – review & editing:** Madeleine Wiebe, Andrew Chan, Jake Hayward.

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
