## [Decision Letter · Decision Letter 0]

13 Feb 2024

PDIG-D-23-00397

Feasibility characteristics of wrist-worn fitness trackers in health status monitoring for post-COVID patients in remote and rural areas

PLOS Digital Health

Dear Dr. Wiebe,

Thank you for submitting your manuscript to PLOS Digital Health. After careful consideration, we feel that it has merit but does not fully meet PLOS Digital Health's publication criteria as it currently stands. Therefore, we invite you to submit a revised version of the manuscript that addresses the points raised during the review process.

Please submit your revised manuscript within 30 days Mar 14 2024 11:59PM. If you will need more time than this to complete your revisions, please reply to this message or contact the journal office at digitalhealth@plos.org. Please include the following items when submitting your revised manuscript:

We look forward to receiving your revised manuscript.

Kind regards,

Martin G Frasch

Section Editor

PLOS Digital Health

Journal Requirements:

Additional Editor Comments (if provided):

I concur with the reviewer's comments and, if addressed, should lead to an acceptance. This is a well written article and contributes to the literature. 

In addition, it is important for the authors to include a few additional limitations. These include 1. Consumer devices are tuned to healthy populations and can give significant errors on patients with medical conditions, and so only relative changes in metrics should be considered. 2. Companies can change algorithms through automatic updates without the user knowing. This can cause distribution shifts that can affect interpretation of results. 3. The proprietary nature of such devices can obscure exactly how metrics are calculated, making comparisons between devices or even different models of devices difficult, if not impossible.

Reviewers' comments:

Reviewer's Responses to Questions

**Comments to the Author**

1. Does this manuscript meet PLOS Digital Health’s publication criteria? Is the manuscript technically sound, and do the data support the conclusions? The manuscript must describe methodologically and ethically rigorous research with conclusions that are appropriately drawn based on the data presented.

Reviewer #1: Yes

Reviewer #2: Yes

2. Has the statistical analysis been performed appropriately and rigorously?

Reviewer #1: Yes

Reviewer #2: Yes

3. Have the authors made all data underlying the findings in their manuscript fully available (please refer to the Data Availability Statement at the start of the manuscript PDF file)?

Reviewer #1: Yes

Reviewer #2: Yes

4. Is the manuscript presented in an intelligible fashion and written in standard English?

Reviewer #1: Yes

Reviewer #2: Yes

5. Review Comments to the Author

Reviewer #1: Overall, the manuscript presents a valuable contribution to the field of post-COVID monitoring using consumer-grade biosensors. It also helps to uncover the impact of the disease in rural settings, and the interplay with health technology solutions, which makes the manuscript further appealing as these cohorts can be under-reported in the literature. Although, as the authors are surely aware, there were clearly significant challenges in reaching a satisfactory sample size and therefore significant limitations in the generalisability of the study’s findings. 

Introduction:

This section clearly outlines the background and motivation for the study, supporting the rationale for using wearable devices in post-COVID conditions. It also highlights the importance of remote patient monitoring, especially in rural areas, and addresses a significant gap in healthcare. Although the introduction describes the feasibility of using FitBits for disease parameter tracking and patient’s experiences, a clearer statement of the research question or hypothesis would enhance reader understanding. 

Methods:

The patient cohort is well described; however, the inclusion and exclusion criteria could be expanded upon:

- Were there minimum and maximum age restrictions? 

- Were pre-existing conditions or co-morbidities considered? 

- Were drug, alcohol, and substance abuse consumption considered?

- Did patients need to have a smartphone to be included? If so, was there any attempt to provide study smartphones or tablets for those who could have been excluded because of this?

Additionally, were specific researchers screening referrals through the ESD programme and virtual post-covid pilot program (PCPP), or were clinicians identifying patients and passing on to the research team? This isn’t entirely clear. Line 126 also only refers to ESD patients being screened for eligibility, was this different if patients were received from the PCPP or through the rehabilitation line? 

Devices: 

It might be worth noting whether FitBit Charge 2 is a medical device, or if there are any limitations? (Irwin & Gary, 2022. PMID: 36711436)

Statistical analysis:

- What statistical program(s) was used to perform the analysis? 

- Was there any assessment to determine the power required for the secondary outcome analysis (e.g. G*Power)?

Finally, the methods section could briefly mention how qualitative and quantitative data will be integrated in the analysis and interpretation stages – a key strength of this study is the holistic understanding of that data for the reader.

Results: 

Participant characteristics:

As noted further in the manuscript, there was clearly a significant challenge to obtain a high recruitment rate. As Canada was leaving its fourth infection wave and entering the fifth during the recruitment period was this clinical population particularly unaffected due to the rural setting? It could be useful to know what percentage of the 21,000 residents presented with COVID complications in the study period, or the overall infection rate of Camrose. 

Whilst this section provides the overall recruitment rate, reasons for refusal and withdrawal, it would be helpful to understand why 3 withdrew consent and the patient with technical difficulties did not have their device replaced. 

Descriptive analysis:

- Line 263. It may be clearer to specify the exact weeks being analysed? E.g. weeks 1-4, weeks 5-8, 8+ etc 

- Line 264. The authors mention the relevant statistical significance for the respiratory symptoms presented in Table 2 but omit any other higher-level analysis completed (although this is presented further in supplementary materials). It would be useful for the reader to see the relevant non-significant, trend, and significance values in the table. 

- Line 280. It would be helpful for the reader if significant values were highlighted. 

- Line 304. Figure 2 indicates individual sensor and longitudinal data between Patient A and B. These are quite hard to read and would benefit with being joined together where possible, for example, overlaying heart rate over steps graphs. Explaining what mental health symptoms were prominent would also be useful. 

The presentation of survey results and thematic analysis provides qualitative insights into patient experiences and perceptions. This qualitative data enriches the overall understanding of the study outcomes. However, was there any consideration for an analysis for those hospitalised vs those who were recruited from community?

Discussion: 

The discussion appropriately considers human factors such as patient motivation, device familiarity, and the importance of convenience in determining the success of remote patient monitoring. The note on machine learning techniques to handle complex and heterogeneous biosensor data is crucial, but it would be beneficial to provide a bit more detail on how these techniques could be applied or developed. 

As noted, the small sample size proves difficult to extrapolate these findings to the wider community and further studies, especially with the onset of vaccines and phenotype variations, however reflecting on the potential selection bias (younger cohort), and the focus on a specific rural population could contribute to a more balanced interpretation of the findings. A key weakness is that those that remained in the study or consented could be those more familiar with wearable technology, indicating perhaps a slight bias towards this subgroup and not necessarily digitally inclusive or representative of the wider clinical environment. Further supporting those who are digitally illiterate would help to maximise the impact and generalisability of any future study.

The conclusion provides a clear summary of the promising characteristics of wrist-worn devices in remote disease tracking for post-COVID conditions in rural communities. This aligns well with the study's outcomes and indicates positive future applications of wearable technologies in clinical settings.

Reviewer #2: The paper is well-written, clear, and concise, providing a valuable contribution to the timely topic of post-COVID medical monitoring in rural communities.

Your introduction highlights the potential utility of common consumer-grade biosensors for post-discharge medical monitoring, particularly in remote areas. The methods section is comprehensive and well-implemented, offering a clear overview of your study design and data collection processes. The results section presents meaningful findings, and the correlations between biosensor data and patient-reported symptoms add depth to the feasibility analysis.

I have only a few minor comments to enhance the manuscript further:

1.It would be beneficial to include additional information regarding the time resolution or frequency at which biosensor parameters, precisely heart rate, were recorded. Understanding whether these metrics are aggregated over a specific time interval (e.g., 30 seconds, 1 minute) to interpret the collected data better would be helpful.

2.Clarify the metric "% of minutes with analyzable heart rate data" in the primary outcome definition as a reflection of adherence. Are there potential technical issues or incorrect positioning that may lead to unreliable data that counts as non-adherence to the protocol?

3.Consider enhancing the quality of Figure 2 (mean resolution) and adding the unit metric to the y-axis, which will significantly improve its usability and contribute to the overall clarity of the presentation.

6. PLOS authors have the option to publish the peer review history of their article (what does this mean?). If published, this will include your full peer review and any attached files.

**Do you want your identity to be public for this peer review?** For information about this choice, including consent withdrawal, please see our Privacy Policy.

Reviewer #1: No

Reviewer #2: No

---

## [Decision Letter · Decision Letter 1]

4 Jul 2024

Feasibility characteristics of wrist-worn fitness trackers in health status monitoring for post-COVID patients in remote and rural areas

PDIG-D-23-00397R1

Dear Miss Wiebe,

We are pleased to inform you that your manuscript 'Feasibility characteristics of wrist-worn fitness trackers in health status monitoring for post-COVID patients in remote and rural areas' has been provisionally accepted for publication in PLOS Digital Health.

Best regards,

Martin G Frasch

Section Editor

PLOS Digital Health

Reviewer Comments (if any, and for reference):

Reviewer's Responses to Questions

**Comments to the Author**

1. If the authors have adequately addressed your comments raised in a previous round of review and you feel that this manuscript is now acceptable for publication, you may indicate that here to bypass the “Comments to the Author” section, enter your conflict of interest statement in the “Confidential to Editor” section, and submit your "Accept" recommendation.

Reviewer #1: All comments have been addressed

2. Does this manuscript meet PLOS Digital Health’s publication criteria? Is the manuscript technically sound, and do the data support the conclusions? The manuscript must describe methodologically and ethically rigorous research with conclusions that are appropriately drawn based on the data presented.

Reviewer #1: Yes

3. Has the statistical analysis been performed appropriately and rigorously?

Reviewer #1: Yes

4. Have the authors made all data underlying the findings in their manuscript fully available (please refer to the Data Availability Statement at the start of the manuscript PDF file)?

Reviewer #1: Yes

5. Is the manuscript presented in an intelligible fashion and written in standard English?

Reviewer #1: Yes

6. Review Comments to the Author

Reviewer #1: The authors have provided a detailed and thorough response to the comments during the review and have sufficiently improved their paper. The manuscript now reads with a greater clarity and focus, adequately highlighting the limitations of their study (and areas of digital health in general) whilst providing insight to a difficult and under reported area of the literature. 

Minor corrections:

- Please remove the repeated PSS acronym from line 170 as this is repeated from line 169. 

- Please remove the bolded r and p values in lines 295 and 296, as you've now bolded these in Table 3 for clarity.

- Line 469. Please change "replicated a larger" to "replicated to a larger".

7. PLOS authors have the option to publish the peer review history of their article (what does this mean?). If published, this will include your full peer review and any attached files.

**Do you want your identity to be public for this peer review?** For information about this choice, including consent withdrawal, please see our Privacy Policy.

Reviewer #1: No
